# The Association between BMI, Days Spent in Hospital, Blood Loss, Surgery Time and Polytrauma Pelvic Fracture—A Retrospective Analysis of 76 Patients

Tomasz Pielak [1], Rafał Wójcicki [1], Piotr Walus [1], Adam Jabłoński [1], Michał Wiciński [2], Przemysław Jasiewicz [3], Bartłomiej Małkowski [4], Szymon Nowak [5] and Jan Zabrzyński [1,5,*]

1. Department of Orthopaedics and Traumatology, Faculty of Medicine, J. Kochanowski University in Kielce, 25-001 Kielce, Poland; tomasz.pielak@gmail.com (T.P.); ralfw@wp.pl (R.W.); walus.md@gmail.com (P.W.); adam.jablonski19941@gmail.com (A.J.)
2. Department of Pharmacology and Therapeutics, Faculty of Medicine, Collegium Medicum in Bydgoszcz, Nicolaus Copernicus University in Toruń, 85-092 Bydgoszcz, Poland; wicinski4@wp.pl
3. Department of Anesthesiology, Faculty of Medicine, Collegium Medicum in Bydgoszcz, Nicolaus Copernicus University in Toruń, 85-092 Bydgoszcz, Poland; przemyslaw.jasiewicz@gmail.com
4. Department of Urology, Oncology Centre Prof. Franciszek Łukaszczyk Memorial Hospital, 85-796 Bydgoszcz, Poland; malkowski.b@gmail.com
5. Department of Orthopaedics and Traumatology, Faculty of Medicine, Collegium Medicum in Bydgoszcz, Nicolaus Copernicus University in Toruń, 85-092 Bydgoszcz, Poland; n.szymon09@gmail.com
* Correspondence: zabrzynski@gmail.com

**Abstract:** Objective: The objective of this study was to investigate the association between BMI, days spent in hospital, blood loss, and surgery time in patients who suffered from isolated pelvic fractures and pelvic fractures with concomitant injuries (polytrauma patients). Methods: This study included 76 consecutive patients who were admitted for pelvic ring fracture surgery between 2017 and 2022. The inclusion criteria were pelvic fractures and indications for operative treatment (LC II and III, APC II and III, and VS). The exclusion criteria were non-operative treatment for pelvic ring fractures, acetabular fractures and fractures requiring primary total hip arthroplasty (THA), and periprosthetic acetabular fractures. Demographic data were collected, including age (in years), sex, type of fracture according to Young–Burgess, date of injury and surgery, surgical approach and stabilization methods, mechanism of trauma, concomitant trauma in other regions, body mass index (BMI), blood transfusions, number of days spent in the hospital, and surgery duration. Results: Patients who suffered from a pelvic ring injury with concomitant injuries had a significantly greater amount of blood units transferred (1.02 units vs. 0.55 units), and the length of hospital stay was also longer compared to the mean results (5.84 days vs. 3.58 days), $p = 0.01$ and $p = 0.001$, respectively. Moreover, patients with a higher BMI had more frequent APC II and APC III fractures ($p = 0.012$). Conclusions: This study demonstrates that polytrauma patients who suffered from pelvic ring injury are, indeed, at risk of blood transfusion in terms of greater units of blood and a longer duration of hospital stay. Moreover, BMI has an impact on pelvic ring fracture morphology. However, there is no doubt that there is an absolute need for further studies and investigations to provide better overall management of polytrauma patients with pelvic fractures.

**Keywords:** pelvic fracture; pelvis; BMI; transfusion; Young–Burgess; trauma; polytrauma

## 1. Introduction

In recent years, an undoubtedly profound improvement has been observed in the diagnostics of patients who have suffered from traumatic injuries due to the accessibility of crucial diagnostic tools such as a computer tomography (CT) scan, for instance [1,2]. Nonetheless, traumatic pelvic fractures remain one of the most life-threatening injuries given their characteristics and the concomitant injuries that frequently occur [3,4]. Despite

the fact that pelvic fractures account for only about 3–8% of all skeletal fractures, mortality rates in specific studies were shown to be as high as 15% [5–7]. Many recent publications that concern pelvic fractures are, in fact, focused on blood loss, survival, and initial management [8–16]. There is no doubt that those studies are important and needed, especially since almost 50% to 60% of deaths in pelvic trauma patients are caused by hypovolemic shock caused by hemorrhage that originates from the pelvis [17,18]. However, a broader perspective on other specific factors would also be beneficial in order to provide better treatment for patients who have suffered pelvic fractures. During our investigation, our research team encountered rather a small number of studies that focused on the characteristics of individuals who had suffered from pelvic fractures, such as BMI, age, or sex. Since we believe it is detrimental to explore these factors, we decided to study our population who had suffered from pelvic ring fracture and investigate whether there were any significant correlations with the type of fracture, blood loss, or length of hospital stay. In order to obtain a better understanding of how specific factors may influence the management of patients with pelvic fractures, our research team decided to analyze the issues mentioned above. The aim of this study was to investigate the association between body mass index (BMI), days spent in hospital, blood loss, and surgery time in patients who suffered from isolated pelvic fractures and pelvic fractures with concomitant injuries (polytrauma patients).

## 2. Material and Methods

### 2.1. General Characteristics

This study included 76 consecutive patients who were admitted for pelvic ring fracture surgery between 2017 and 2022. The data were collected prospectively at a single trauma center. All patients underwent operative treatment using De Puy Synthes implants for pelvic fixation. Pelvic fractures were classified according to the Young–Burgess system (LC—lateral compression, APC—anterior–posterior compression, VS—vertical shear). The inclusion criteria were pelvic fractures and indications for operative treatment (LC II and III, APC II and III, VS). The exclusion criteria were non-operative treatment for pelvic ring fractures, acetabular fractures and fractures requiring primary total hip arthroplasty (THA), and periprosthetic acetabular fractures. Upon admission, all patients underwent evaluation using X-ray and computed tomography scans of the pelvis. Demographic data were collected, including age (in years), sex, type of fracture according to Young–Burgess, date of injury and surgery, surgical approach and stabilization methods, mechanism of trauma, concomitant trauma in other regions, BMI, blood transfusions, number of days spent in the hospital, and surgery duration. The surgical approaches used for pelvic fractures were the ilio-inguinal and Kocher–Langenbeck approaches. The mechanism of trauma was categorized according to typical pattern of injury, such as: fall from height (injury to a person that occurs after landing on the ground after falling from a higher place, higher than the human body, very-high-energy trauma), industrial trauma (caused by an accident at work, usually with machines, high-energy trauma), normal fall (suddenly falling down onto the ground or toward the ground without intending to or by accident, from the level of human body, low-energy trauma), pedestrian injury (injury to pedestrians who were struck by motor vehicles, high-energy trauma), traffic accident (an accident involving at least one vehicle on a road open to public traffic in which at least one person is injured, usually a very-high-energy trauma), and unknown mechanism of trauma.

The Young–Burgess classification system is most commonly used for evaluating pelvic ring injuries [19]. In 1986, Young et al. described 142 patients with pelvic ring injuries and classified their injuries based on the direction and location of applied force [20].

Antero–posterior compression (APC) injuries are caused by anterior to posterior-directed force, resulting in a predictable pattern based on the disruption of anterior and posterior pelvic structures, including the symphysis pubis and sacroiliac joints [19]. APC I injuries are characterized by less than 2.5 cm of symphyseal widening and have no posterior instability, either clinically or radiographically. APC II injuries show symphysis pubis widening and instability of the posterior pelvis resulting from disruption of the anterior

sacroiliac complex. APC III injuries are associated with complete posterior ligamentous disruption and have the highest rate of mortality, blood loss, and need for transfusion of all pelvic ring injuries.

Lateral compression (LC) injuries result when a laterally based force directed medially is applied to the pelvis. LC I injuries result from a lateral force delivered over the posterior aspect of the pelvis and represent a spectrum of injuries [19]. LC II injuries result from a more anteriorly directed force, causing internal rotation of the anterior hemipelvis with possible external rotation of the posterior hemipelvis, with the anterior sacroiliac joint serving as a fulcrum. The resulting posterior pelvic injury in LC II patterns may be a sacral fracture, sacroiliac ligament and joint disruption, or crescent fracture–dislocation of the ilium. LC III injuries result from greater force. The rotation of the ipsilateral hemipelvis causes the injury to the contralateral hemipelvis, resulting in LC III.

Vertical shear (VS) injuries result from an axially loaded force delivered over one or both hemipelves lateral to the midline. The sacrum is driven down relative to the iliac wing, resulting in complete ligamentous injury and disruption of the sacrospinous, sacrotuberous, anterior, and posterior sacroiliac ligaments on the injured side [19].

Complex injury patterns are a combination of any three primary patterns (APC, LC, or VS), usually resulting from LC injuries mixed with AP or VS trauma.

### 2.2. Ethics

The study was conducted in accordance with the Declaration of Helsinki guidelines for human experiments. Prior to the study, permission was obtained from the local Bioethics Committee (approval number KB 645/2022). Written informed consent was obtained from all patients or their relatives upon admission to the hospital to include them in scientific studies.

### 2.3. Statistical Analysis

All group comparisons and statistical analyses were conducted by two independent investigators using Prism software (GraphPad 8). A $p$-value of less than 0.05 was considered statistically significant. Nominal variables were described by the number of observations and their distribution. Normality of variables was assessed using the Shapiro–Wilk test. Relationships between the studied parameters were evaluated using the Spearman's rank correlation coefficient. Non-parametric tests, such as the Mann–Whitney U test and analysis of variance, were used to compare the data.

### 3. Results

A total of 76 patients who underwent operative treatment for pelvic ring fractures between 2017 and 2022 met the inclusion criteria and were included in the study (Chart 1). The inclusion criteria required patients to have a pelvic fracture demanding surgery, either with or without concomitant injury.

The studied cohort had a mean BMI of 26.4 (ranging from 17.8 to 44.8). The mean duration of surgery was 102.4 min, ranging from 35 to 270 min. The average number of blood transfusions received was 0.8 units, in a range of 0 to 5 units. The mean length of hospital stay was 4.9 days, ranging from 1 to 36 days. In total, 23 patients (%) were female and 53 patients (%) were male. Statistical analysis showed significant differences between male and female populations in terms of BMI ($p = 0.01223$), (Figure 1A) but none in duration of surgery ($p = 0.8104$), blood transfusion ($p = 0.8066$), and length of hospital stay ($p = 0.5628$) (Figure 1B–D).

In the female subgroup, the mean BMI was 24.36 (ranging from 17.85 to 33.33), while in the male subgroup, it was 27.25 (ranging from 17.9 to 44.87). The average surgery duration for females was 95.91 min (ranging from 25 to 240 min), and for males, it was 105.1 min (ranging from 35 to 260 min). The mean length of hospital stay for females was 4.1 days (ranging from 2 to 23 days), whereas for males, it was 5.34 days (ranging from 1 to 36 days).

Additionally, the mean amount of blood transfused for females was 0.73 units (ranging from 0 to 5 units), while for males, it was 0.88 units (ranging from 0 to 4 units).

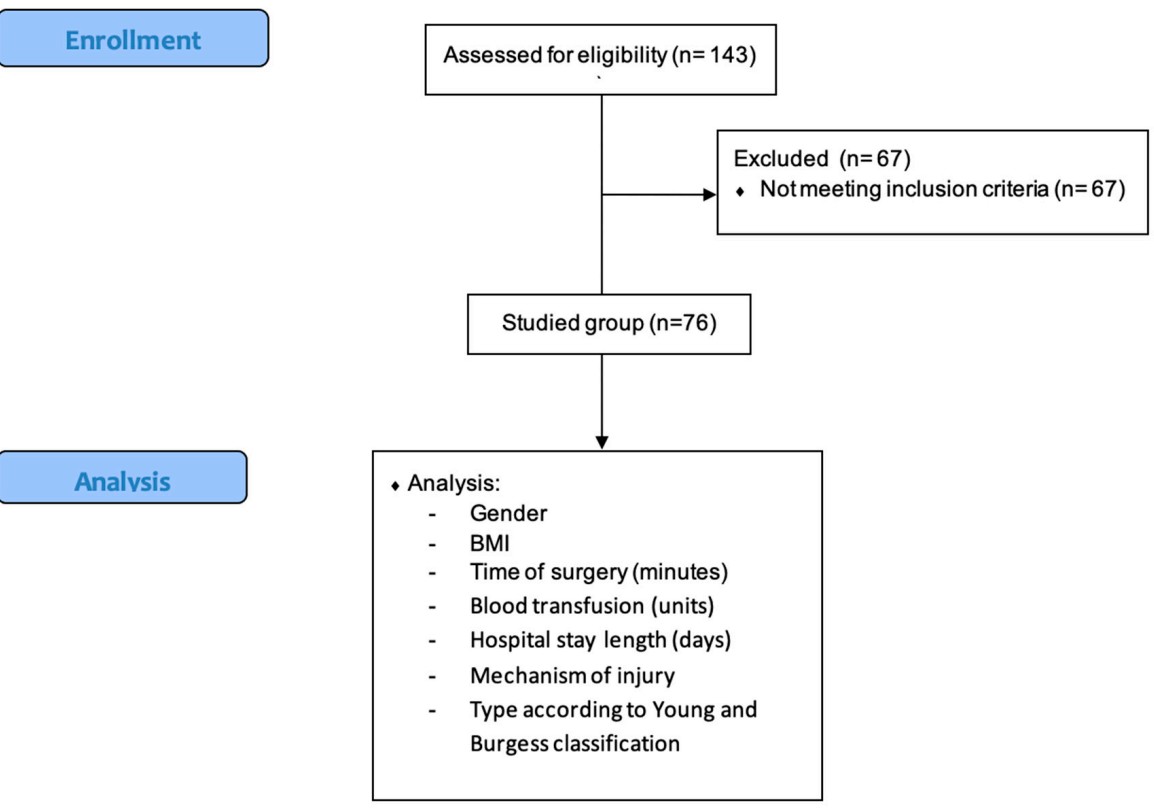

**Chart 1.** A flow diagram of patients included in the study and interventions.

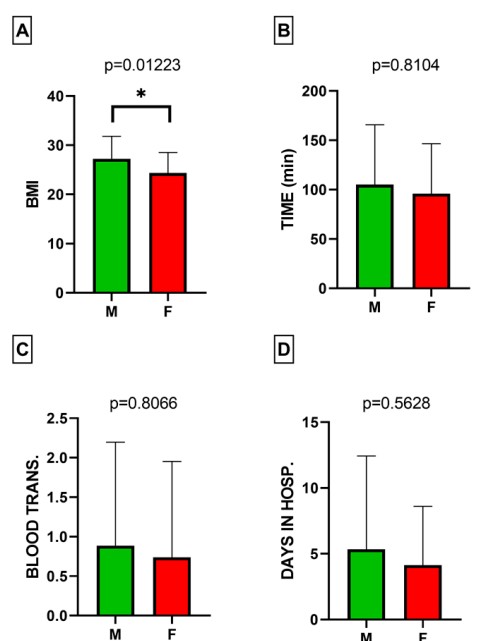

**Figure 1.** (**A**) Comparison of BMI between male and female subgroups. (**B**) Comparison of surgery duration between male and female subgroups. (**C**) Comparison of blood transfusion between male and female subgroups. (**D**) Comparison of length of hospital stay between male and female subgroups. * *p*-value < 0.005.

When analyzing the population according to the Young and Burgees classification (Table 1), statistically significant differences were observed in BMI ($p = 0.01$) distribution among the subgroups with certain types of fracture (Figure 2A). Patients with a higher BMI had more frequent APC II and APC III fractures. On the other hand, there were no significant differences in the time of surgery ($p = 0.06$), length of hospital stay ($p = 0.12$), or blood transfusion ($p = 0.75$) within certain subgroups (Figure 2B–D).

**Table 1.** Comparison of mean blood transfusion, BMI, surgery time, and length of hospital stay in patients with pelvic fracture according to the Young and Burgess classification.

| YB (Type) | APC II | APC III | LC II | LC III | VS |
|---|---|---|---|---|---|
| Mean Blood transfusion (units) | 0.76 | 1.12 | 0.73 | 1.5 | 0.71 |
| Mean BMI (kg/m$^2$) | 28.94 | 29.31 | 25.38 | 23.87 | 26.34 |
| Mean Time of surgery (min) | 109.60 | 130.70 | 91.07 | 154.20 | 84.29 |
| Mean Length of hospital stay (days) | 3.07 | 9.37 | 4.31 | 8.83 | 4.01 |

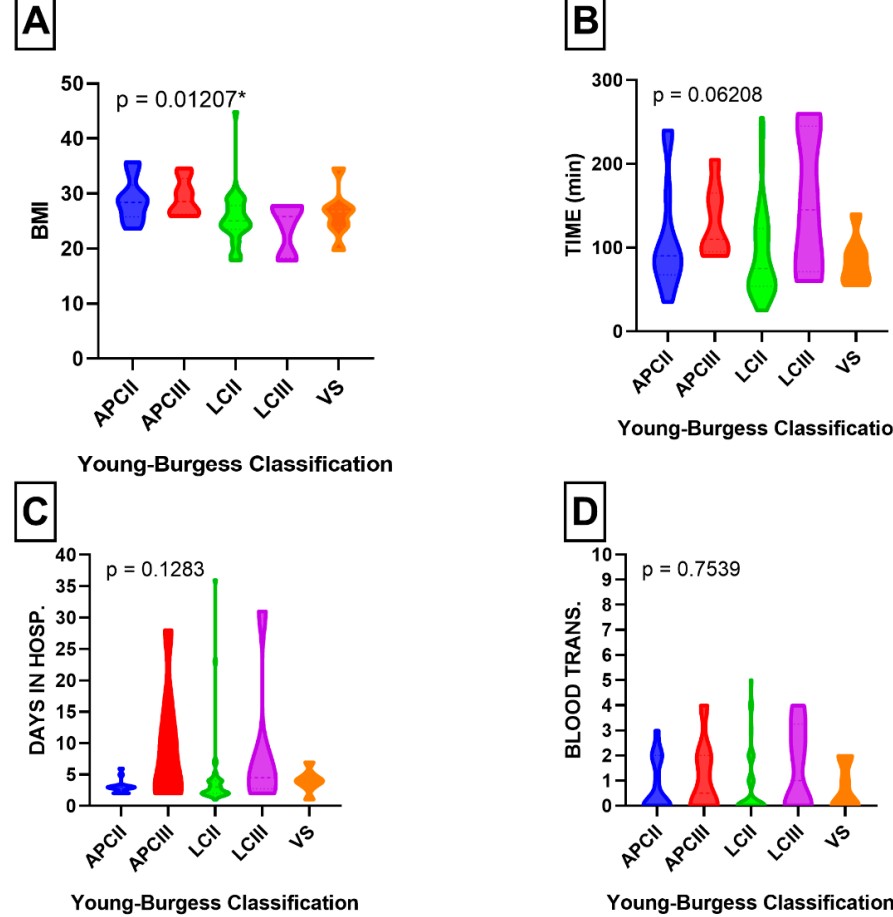

**Figure 2.** (**A**) Comparison of BMI according to Y–B subdivisions. (**B**) Comparison of surgery duration according to Y–B subdivisions. (**C**) Comparison of length of hospital stay according to Y–B subdivisions. (**D**) Comparison of blood transfusion according to Y–B subdivisions. * *p*-value < 0.005.

The Spearman's rho correlation analysis between BMI and surgery duration ($p = 0.44$), blood transfusion ($p = 0.84$), and length of hospital stay ($p = 0.12$) did not reveal a statistically significant relationship (Figure 3A–C).

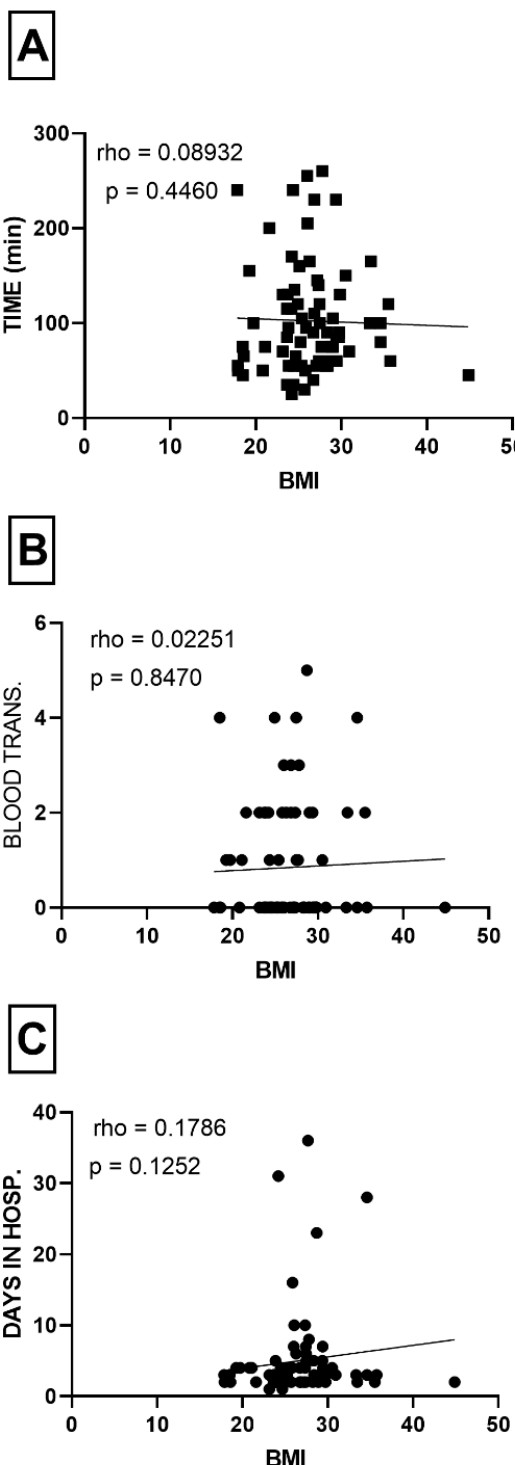

**Figure 3.** (**A**) Correlation between surgery duration and BMI. (**B**) Correlation between blood transfusion and BMI. (**C**) Correlation between length of hospital stay and BMI.

The mechanism of injury exhibited a diverse range, including the following categories: 27 patients experienced falls from height, 2 patients were involved in industrial accidents, 5 patients experienced falls from standing height, 6 patients had pedestrian injuries, 29 patients were involved in traffic accidents, and 7 patients had an unknown mechanism of injury (Table 2).

**Table 2.** Comparison of mean blood transfusion, BMI, surgery time, and length of hospital stay in patients with pelvic fracture in regard to the injury pattern.

| Injury Pattern | Fall from Height | Industrial | Normal | Pedestrian | Car | Unknown |
|---|---|---|---|---|---|---|
| Blood transfusion (units) | 0.81 | 1.5 | 0.4 | 0.83 | 0.75 | 1.42 |
| Mean BMI (kg/m$^2$) | 26.83 | 28.76 | 23.94 | 23.77 | 26.38 | 27.84 |
| Mean Time of surgery (min) | 96.67 | 172.5 | 85 | 130.8 | 98.93 | 106.4 |
| Length of hospital stay (days) | 4.11 | 5.02 | 2.05 | 3.66 | 7.03 | 2.85 |

Classification based on the mechanism of injury did not reveal any statistically significant differences when analyzing BMI ($p = 0.33$), length of hospital stay ($p = 0.055$), surgery duration ($p = 0.30$), and blood transfusion ($p = 0.71$) (Figure 4A–D).

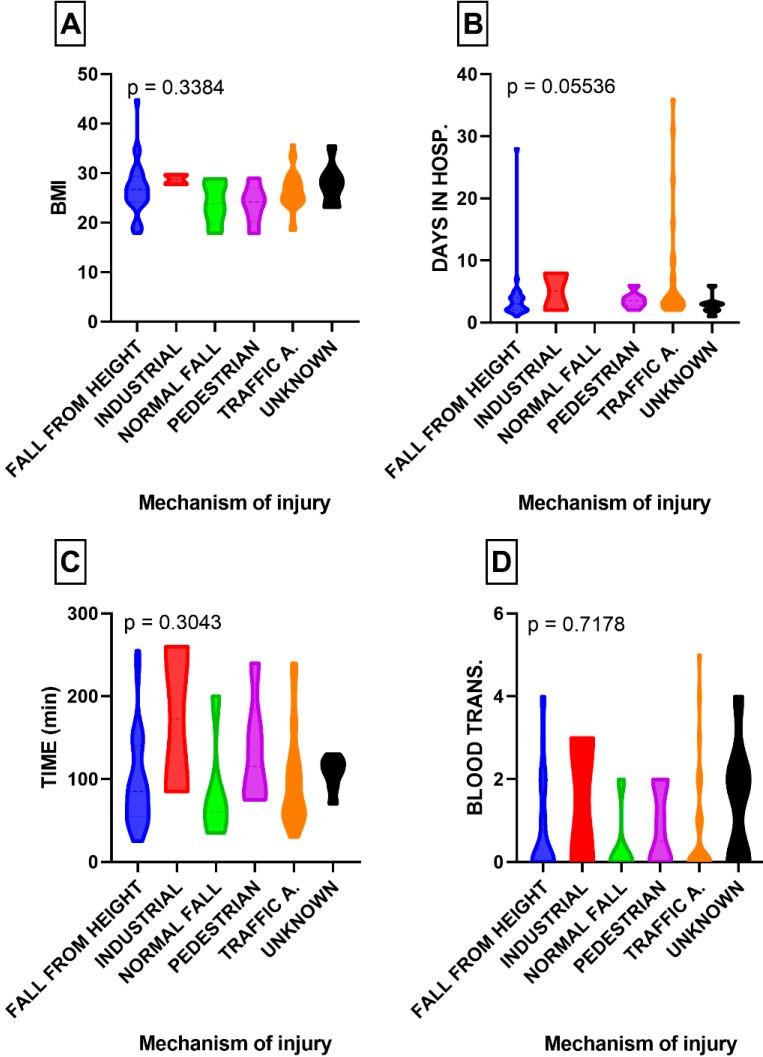

**Figure 4.** (**A**) Comparison of BMI in various mechanism of injury subgroups. (**B**) Comparison of length of hospital stay in various mechanism of injury subgroups. (**C**) Comparison of surgery duration in various mechanism of injury subgroups. (**D**) Comparison of blood transfusion in various mechanism of injury subgroups.

Within the studied cohort, 47 patients had concomitant injuries and were classified as polytrauma patients. When analyzing the relationship between pelvic fractures and additional injuries involving the spine, head, chest, abdomen, and upper/lower limbs, no significant differences were found in terms of BMI ($p = 0.43$) and surgery duration ($p = 0.68$) (Figure 5A,D). Contrary to this, the blood transfusion ($p = 0.01$) and length of hospital stay ($p = 0.001$) were significantly increased in polytrauma patients (Figure 5B,C).

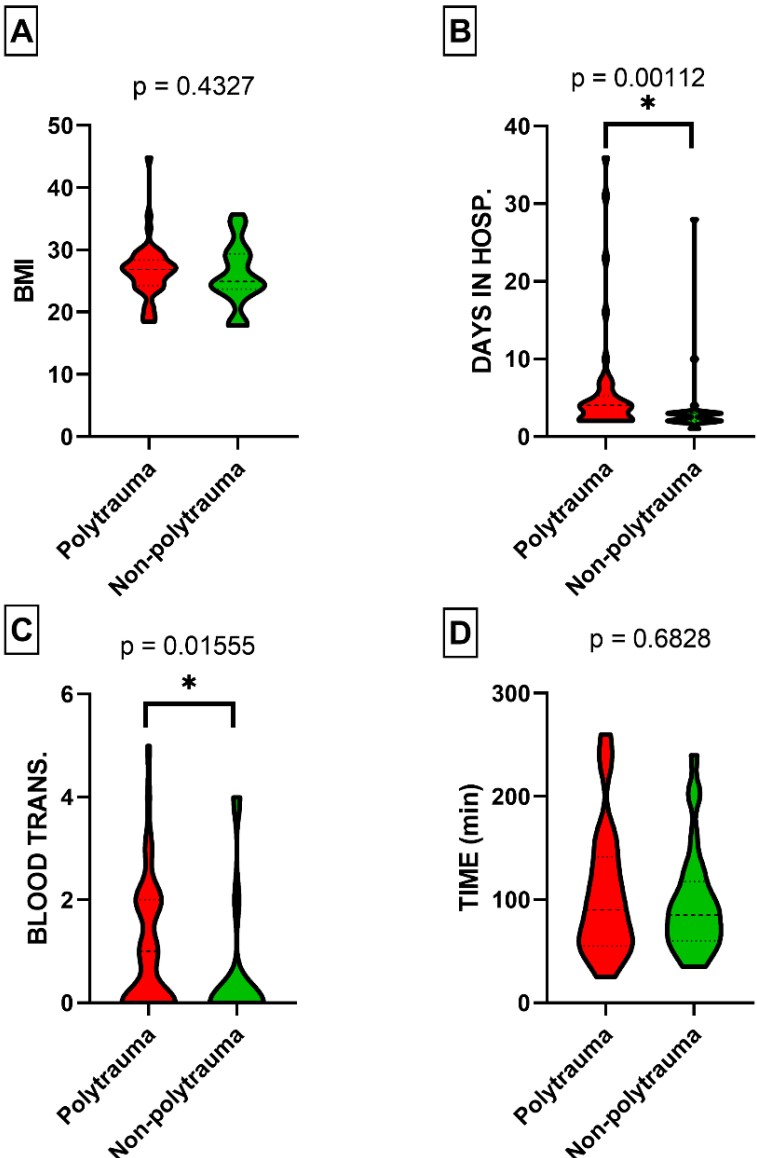

**Figure 5.** (**A**) Comparison of BMI in pelvic fractures with or without polytrauma. (**B**) Comparison of length of hospital stay in pelvic fractures with or without polytrauma. (**C**) Comparison of blood transfusion in pelvic fractures with or without polytrauma. (**D**) Comparison of surgery duration in pelvic fractures with or without polytrauma. * $p$-value < 0.005.

## 4. Discussion

The results of our study demonstrate that patients who suffered a pelvic ring injury with concomitant injuries had significantly greater amounts of blood units transferred, and the length of their hospital stay was also longer compared to the mean results. Both of the provided outcomes were found to be statistically significant. Moreover, the study showed that BMI has an impact on pelvic ring fracture morphology.

In our study, we showed that 70% of patients who suffered from pelvic injury were male and 30% were female; these results were in line with many current studies concerning traumatic pelvic injuries. Abdelrahman et al. showed that 76.5% of the studied population that suffered from traumatic pelvic injury were males [21]. Coleman et al., in their 2020 study on pelvic ring injuries and their association with the Young–Burgess classification, also presented similar conclusions as far as the sex of the patients was concerned. In total, 61% percent of the studied population were male [22]. Another study with a male sample dominance was Veith et al.'s 2016 study, where 60% of patients who suffered from pelvic fractures were male. Ghosh et al. and Cuthbert et al. provided similar results concerning the prevalence of pelvic fractures with regard to sex. Males accounted for 75% and 72% of the studied population with pelvic fractures, respectively [23,24].

Regarding blood transfusions, Magnussen et al. statistically proved, in their study, that patients who suffered from pelvic ring fractures required the highest amount of blood compared to other pelvic fractures [25]. Surprisingly, comparing our results to Magnussen et al., the mean value of blood units transferred was 1.0 in our facility, whereas in Magnussen et al., the value was 5.03 [25]. The difference is large, undoubtedly. However, we suspect this may be connected to the regional requirements and standards in blood transfusions. In Magnussen et al.'s study, there is no description of transfusion criteria. In our study, patients with a hemoglobin level < 7.0 mg/dL or with anemia symptoms qualified to receive a blood transfusion.

Another finding concerning our study is that APC III fractures were found to have a 23% higher amount of blood units transferred compared to APC II-type fractures (0.8 units vs. 1.1 units). Additionally, LC III fractures were shown to require over 105% of blood units transferred (0.7 units vs. 1.5 units) compared to LC II fractures. Veith et al. presented, in their study, that the level of free blood volume found during the CT scan increases with the level of instability of the pelvic ring fracture in the AO/OTA C-types [26]. Not only does this support our findings, but it also indicates the importance of a thorough initial examination in order to establish the injury and associated injuries. Having said that, it is essential to acknowledge that the greater the displacement of the pelvic ring fracture, the higher the chance of severe bleeding and associated abdominal injuries [22,27]. The severity of the bleeding increasing with the instability of the pelvis is believed to be associated with the disruption of the sacral venous plexus and arterial branches of iliac vessels and the high amount of cancellous bone [26,27]. The presented results are in line with current knowledge that pelvic ring fractures can be mortal injuries that may require rapid intervention [14,28–30].

The findings regarding the Young and Burgess classification and its correlation to BMI are as follows: patients who suffered from an APC III injury had a higher mean BMI than the APC II injury patients (28.9 vs. 29.3). This finding correlates with Waseem et al.'s study, where BMI was investigated in regard to pelvic fractures and their outcomes. They demonstrated that the highest number of APC type III injuries were found in the most obese patients [31]. Interestingly, these findings did not correlate to the lateral compression injuries in our study. In fact, we demonstrated that the LC II injury patients had a higher mean BMI than the LC III injury patients (25.4 vs. 23.9). Therefore, it is presented that only in an APC-type pelvic injury does an increasing BMI correlate with the severity of the trauma. Regarding the time of surgery, despite there being no statistically significant differences between Young and Burgess type and degrees of fractures, it should be noted that the LC II-type injury had the lowest mean time in surgery. The data from this study suggest that this was near the statistically significant cut-off ($p$ = 0.0620). Thus, it should be noted that patients with APC III- and LC III-type fractures demand more time spent in the operation theater.

At this point, it is essential to remember that our study and the studies presented above concern only traumatic pelvic injury, and male predominance seems to correlate only to high-energy trauma. In 2016, Buller et al. presented a study on the analysis of pelvic ring fractures in a US population. The studied group comprised almost 15,000,000

cases registered between 1990 and 2007 [32]. Their results showed that almost 70% of the registered pelvic ring fractures concerned females [32]. This study included every type of pelvic ring fracture, including low-energy fractures that, in many cases, concern patients with osteoporosis [33,34]. However, only 7.7% of patients in the studied population were diagnosed and treated with osteoporosis [32]. As Salari et al. presented in their 2021 worldwide meta-analysis comprising 86 studies, the global female prevalence for osteoporosis is 23.1% [35]. Therefore, we suspect that many of the females who suffered from low-energy pelvic ring fractures had already been suffering from undiagnosed or untreated osteoporosis. Only about 10% of the studied cases involved any operative treatment [32]. Since the majority of high-energy pelvic fractures are treated operatively, we suspect that the vast majority of those fractures in the US population were low-energy fractures. Unfortunately, this study does not include the mechanism of injury; however, we strongly suspect that the leading cause was low-energy falls, since over 50% of all registered cases concerned patients older than 75 years [32].

A similar result to the current study was obtained by our research team with regard to the mechanism of injury and pelvic fracture. Cuthbert et al., in 2022, investigated pelvic fractures that were treated in their trauma center and also managed to present the mechanism of injury associated with a specific type of pelvic fracture. They presented that 43% of all pelvic ring fractures in the studied population occurred due to a fall from height, and this was found to be the leading cause of pelvic ring fractures [23]. In our study, we discovered that falls from heights were the second leading cause of pelvic ring fractures at 36%. The most common type of mechanism that resulted in pelvic ring injury was observed to be car accidents at 38.6%. Interestingly, Cuthbert et al. demonstrated that car accidents are responsible for only 13% [23]. Abdelrahman et al. came to a similar conclusion concerning the frequency of the mechanism of injury. In total, 33.6% of the studied population that suffered a pelvic fracture sustained it due to a fall from height. Therefore, our leading cause of pelvic ring fractures appears to correlate with the recent literature.

As stated initially in the discussion, our study proved that, within our studied group, patients with polytraumatic injuries received a greater amount of blood units and had longer hospital stays compared to the non-polytrauma patients. These results were proven to be statistically significant, with $p = 0.01$ and $p = 0.001$ for blood transfusion units and days spent in hospital, respectively. The polytrauma patients received almost 60% more blood units compared to patients who sustained isolated pelvic ring fractures (mean value 1.0 vs. 0.6 units of blood). There is no doubt that these findings were to be expected due to the nature of polytrauma injuries involving pelvic fractures, which, in many cases, are characterized by massive blood loss and an urgent need for blood transfusions [22,36–38]. In our study, a total of 62% of patients were found to suffer additional injuries with a pelvic ring fracture. Our findings agree with a recent study by Coleman et al. They discovered that 69% of the patients who sustained a pelvic ring injury also had concomitant injuries [22]. Magnussen et al. investigated isolated pelvic and isolated acetabular fractures. They demonstrated that 38% of pelvic ring fractures are not associated with concomitant injuries, regardless of the mechanism of injury [25]. Interestingly, the percentage of isolated pelvic ring fractures in our studied population is exactly the same, equal to 38% of all pelvic ring fractures.

What also may not come as a surprise is that the patients who sustained concomitant injuries with pelvic ring fractures had an over 62% longer stay in the hospital compared to the patients who suffered an isolated pelvic ring fracture (mean 5.8 days vs. 3.6 days). The reason for this is obviously the overall worse initial condition of the patients with additional injuries, greater blood loss, and the further need for eventual surgeries concerning other injuries, such as fractures of the long bones, thoracic or abdominal organs, spine or head trauma [39,40]. Our results also seem to be coherent with the 2012 study of Holstein et al. They presented a study on the predictors of mortality in traumatic pelvic fractures. They discovered that patients who did not survive were characterized by severe multiple trauma, massive blood loss, and being of the male gender [39]. All of the above-mentioned cate-

gories characterize our dominant group of traumatic pelvic injuries. Therefore, it seems rational to suggest that males who sustained pelvic ring fractures and concomitant injuries should be carefully and thoroughly examined since they have the highest statistical chance of death.

Nonetheless, it is important to point out the limitations of our study. One of the limitations is the sample size, primarily consisting of a local population, represented by a single trauma center. As a retrospective study, there may be biases that could influence our results and attitudes towards the outcomes. Another issue concerns the operative team, which only consisted of two main operators. Not only did they perform all of the procedures but they also assisted each other during their surgeries. We believe that a greater number of operators would provide less biased results and could provide a broader perspective on the issue of pelvic fractures and their associated issues.

## 5. Conclusions

Specific types of pelvic fracture, according to the Young and Burgess classification, have not been linked to blood loss, duration of hospital stay, or concomitant injuries. This study demonstrates that polytrauma patients who have suffered a pelvic ring injury are, indeed, at risk of a greater amount of blood units needed during a transfusion and a longer duration of hospital stay. A higher BMI was more frequently associated with APC III- and APC II-type fractures in the Young and Burgess classification. There is an absolute need for further studies and investigations in this area to provide better overall management of polytrauma patients with pelvic fractures.

**Author Contributions:** Conceptualization, T.P. and J.Z.; methodology, J.Z. and M.W.; software, A.J. and B.M.; validation, R.W. and T.P; formal analysis P.W. and J.Z.; investigation P.W., A.J., J.Z., T.P. and P.J.; resources, T.P., R.W. and S.N.; data curation B.M., M.W. and S.N.; writing original draft preparation, T.P., J.Z. and P.W.; writing—review and editing, J.Z. and P.W., visualization, A.J., S.N. and P.J.; supervision, T.P., J.Z. and R.W.; project administration; T.P., J.Z. and P.W. All authors have read and agreed to the published version of the manuscript.

**Funding:** The research received no external funding.

**Institutional Review Board Statement:** The study was conducted in accordance with the Declaration of Helsinki. Prior to the study, permission was obtained from the local Bioethics Committee Nicolaus Copernicus University in Torun (approval number KB 645/2022).

**Informed Consent Statement:** Informed consent was obtained from all subjects involved in the study.

**Data Availability Statement:** The data is available upon a specific request.

**Conflicts of Interest:** The authors declare no conflict of interest.

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
