# Peer review of "The Association between BMI, Days Spent in Hospital, Blood Loss, Surgery Time and Polytrauma Pelvic Fracture—A Retrospective Analysis of 76 Patients"

_applsci, doi:10.3390/app131810546_

Round 1
Reviewer 1 Report
A manuscript with a very interesting topic, well written and edited.
Add a CONSORT type scheme to make the presentation clearer.
Clinical applicability and limitations of the study should be added.
Author Response
Thank you for the opportunity to improve and resubmit our manuscript entitled:
“The association between BMI, days spent in hospital, blood loss, surgery time and polytrauma pelvic fracture – a retrospective analysis of 76 patients”
The suggestions offered by the reviewers have been immensely helpful. We appreciate all the comments on the manuscript.
We have included the reviewer comments, and responded to them individually, indicating how we addressed each concern and describing the changes we have made:
Reviewer 1:
A manuscript with a very interesting topic, well written and edited.
Add a CONSORT type scheme to make the presentation clearer.
Clinical applicability and limitations of the study should be added.
Answers:
- The flow diagram was added.
- Limitations are in the end of the discussion section.
- Clinical applicability was added to introduction section.
Reviewer 2 Report
The authors have made an interesting attempt at “The Association between BMI, Days Spent in Hospital, Blood Loss, Surgery Time and Polytrauma Pelvic Fracture – A Retro- Spective Analysis of 76 Patients.” The manuscript is interesting; however, the authors need to justify the scientific writing manuscript. Some of the general comments are provided below:
1. Is the sample size of 76 patients adequate to draw meaningful conclusions about the studied population of pelvic ring fracture patients?
2. Were there any specific criteria for selecting these 76 consecutive patients? Were there any potential biases in patient selection?
3. Can you provide more details about the specific De Puy Synthes implants used for pelvic fixation? Were there variations in implant types among the patients?
4. The range of surgery duration appears to be quite wide (35 to 270 minutes). Were there any outliers or extreme values that might have influenced the mean duration significantly?
5. The difference in mean BMI between the female and male subgroups is statistically significant (p=0.01). Were there any specific reasons or medical explanations for this difference?
6. How were the diverse mechanisms of injury categorized and their impact on the outcomes assessed? Did the mechanism of injury correlate with the type or severity of pelvic fractures?
7. How might the variations in surgery duration, blood transfusion, and length of hospital stay impact patient recovery, postoperative complications, and long-term outcomes?
8. Were there any changes in trauma care protocols or practices during the study period (2017-2022) that could influence patient outcomes, especially considering the diverse mechanisms of injury?
Author Response
Thank you for the opportunity to improve and resubmit our manuscript entitled:
“The association between BMI, days spent in hospital, blood loss, surgery time and polytrauma pelvic fracture – a retrospective analysis of 76 patients”
The suggestions offered by the reviewers have been immensely helpful. We appreciate all the comments on the manuscript.
We have included the reviewer comments, and responded to them individually, indicating how we addressed each concern and describing the changes we have made:
Reviewer 2:
- Is the sample size of 76 patients adequate to draw meaningful conclusions about the studied population of pelvic ring fracture patients?
- Were there any specific criteria for selecting these 76 consecutive patients? Were there any potential biases in patient selection?
- Can you provide more details about the specific De Puy Synthes implants used for pelvic fixation? Were there variations in implant types among the patients?
- The range of surgery duration appears to be quite wide (35 to 270 minutes). Were there any outliers or extreme values that might have influenced the mean duration significantly?
- The difference in mean BMI between the female and male subgroups is statistically significant (p=0.01). Were there any specific reasons or medical explanations for this difference?
- How were the diverse mechanisms of injury categorized and their impact on the outcomes assessed? Did the mechanism of injury correlate with the type or severity of pelvic fractures?
- How might the variations in surgery duration, blood transfusion, and length of hospital stay impact patient recovery, postoperative complications, and long-term outcomes?
- Were there any changes in trauma care protocols or practices during the study period (2017-2022) that could influence patient outcomes, especially considering the diverse mechanisms of injury?
Answers:
- Pelvic fractures are rather treated in distinguished and dedicated orthopedic & traumatology centers. To obtain an adequate population is difficult to express, because these patients are usually limited. In our opinion 76 patients with pelvic ring fractures, which demands operative treatment according to Y&B classification, without acetabulum fracture, is an adequate sample size. This study was conducted in one trauma center, surgeons who did the surgery were the same in each case, and this is important from scientific point of view.
- We used clear inclusion and exclusion criteria.
- De Puy Synthes implants used in the studied population was 3.5 mm Low Profile Pelvic System, dedicated System for Reconstructive Pelvic and Acetabular Surgery.
- The LC II fractures according to Y&B classification had lower time of surgery due to, usually, less complicated fracture morphology. We also added to discussion:
“Regarding the time of surgery, despite there was no statistically significant differences between Young and Burgess type and degrees of fractures, it should be noted that, the LC II type of injury had the lowest mean time of surgery. The data from this study suggest that it was near statistically significant cut-off (p=0.0620). Thus it should be noted, that patients with APC III and LC III types of fractures demand more time spent in the operation theater.”
- There is no clear explanation for this fact. Perhaps the data in our country about the rising BMI values in females could an explanation.
- A specific explanation was added in the M&M section:
“The mechanism of trauma was categorized according to typical pattern of injury, such as: fall from height (injury to a person that occurs after landing on the ground after falling from a higher place, higher than human body, very high energy trauma), industrial trauma (caused by an accident at work, usually with machines, high energy trauma), normal fall (suddenly go down onto the ground or toward the ground without intending to or by accident, from the level of human body, low energy trauma), pedestrian injury (injury to pedestrians who were struck by motor vehicles, high energy trauma), traffic accident (an accident involving at least one vehicle on a road open to public traffic in which at least one person is injured, usually a very high energy trauma), unknown mechanism of trauma.”
- This question will be answered in the next papers.
- The surgeons team is still the same since 2017. There were no alterations in treatment protocols.
Reviewer 3 Report
The article is well described, however, the novelty of this article is low. Authors should emphasize the originality of current research.
Specific comments
- The introduction does not provide a sufficient background and rationale for the study. It should include a review of the relevant literature, a clear statement of the research question and hypothesis, and an explanation of the significance and novelty of the study.
- How did you define the indications for operative treatment of pelvic fractures? Did they use Young-Burgess system only? If so, please describe the details of the system.
- Figure 1 ; Graphs should have an error bar. In Fig 1D, stay in the hospital seemed longer than described in the text.
- Please discuss why only APC type fractures showed a relationship between BMI. If it is a sample size matter, please state.
Author Response
Thank you for the opportunity to improve and resubmit our manuscript entitled:
“The association between BMI, days spent in hospital, blood loss, surgery time and polytrauma pelvic fracture – a retrospective analysis of 76 patients”
The suggestions offered by the reviewers have been immensely helpful. We appreciate all the comments on the manuscript.
We have included the reviewer comments, and responded to them individually, indicating how we addressed each concern and describing the changes we have made:
Reviewer 3
Specific comments
- The introduction does not provide a sufficient background and rationale for the study. It should include a review of the relevant literature, a clear statement of the research question and hypothesis, and an explanation of the significance and novelty of the study.
- How did you define the indications for operative treatment of pelvic fractures? Did they use Young-Burgess system only? If so, please describe the details of the system.
- Figure 1 ; Graphs should have an error bar. In Fig 1D, stay in the hospital seemed longer than described in the text.
- Please discuss why only APC type fractures showed a relationship between BMI. If it is a sample size matter, please state.
- The introduction section was extensively revised.
- The Y&B systems was extensively explained in the M&M section.
- The Figure 1 was checked and edited.
- It is explained in the discussion section:
“Findings regarding the Young and Burgess classification and its correlation to BMI are as follows: patients who suffered from APC III injury had higher mean BMI that APC II injury patients (28.9 vs 29.3). This finding correlates with Waseem et al. study where BMI was investigated in regards to pelvic fractures and its outcomes. They presented that the highest number of APC type III injuries were found in the most obese patients [21]. Interestingly, those findings where not correlating to lateral compression injuries in our study. In fact, we found that LC II injury patients had higher mean BMI than LC III injury patients (25.4 vs 23.9). Therefore, it has been presented that only in APC type of pelvic injury, increasing BMI correlates in the severity of the trauma. “
Round 2
Reviewer 2 Report
The authors have addressed my queries and now the manuscript is acceptable for publication.